# DNA Damage Response and Immune Defense

**DOI:** 10.3390/ijms21207504

**Published:** 2020-10-12

**Authors:** Claudia Nastasi, Laura Mannarino, Maurizio D’Incalci

**Affiliations:** Department of Oncology, Istituto di Ricerche Farmacologiche Mario Negri IRCCS, Via Mario Negri 2, 20156 Milan, Italy; laura.mannarino@marionegri.it

**Keywords:** DNA damage response, DNA repair, immune defense, immune signalling, innate immunity, cancer

## Abstract

DNA damage is the cause of numerous human pathologies including cancer, premature aging, and chronic inflammatory conditions. The DNA damage response (DDR), in turn, coordinates DNA damage checkpoint activation and promotes the removal of DNA lesions. In recent years, several studies have shown how the DDR and the immune system are tightly connected, revealing an important crosstalk between the two of them. This interesting interplay has opened up new perspectives in clinical studies for immunological diseases as well as for cancer treatment. In this review, we provide an overview, from cellular to molecular pathways, on how DDR and the immune system communicate and share the crucial commitment of maintaining the genomic fitness.

## 1. Introduction

Genomic integrity and stability are the center upon which cellular survival and homeostasis stand. In dividing human cells, the entire genome is replicated every few hours [1]. Findings in the first half of the 20th century demonstrated that DNA could be altered as well as damaged. However, the notion of DNA repair did not become a common term in the field of molecular and cellular biology until the 1960s [2]. Every cell in the body employs distinct but interrelated systems to detect and eliminate DNA damage so that deleterious errors can be avoided by accurate replication [3]. The awareness of how DNA repair pathways could be targeted to kill cancer cells came many steps and many discoveries later [2,4,5]. However, we know that the relationship between DNA damage and several pathologies, in primis cancer, is undeniable.

In fact, human DNA is exposed daily to a variety of exogenous and endogenous genotoxic insults, which can result in DNA damage. During DNA synthesis, the replication machinery must overcome numerous obstacles, such as lesions that interfere with fork progression, tightly bound DNA-protein complexes, and non-B form DNA structures (cruciforms, slipped structures, triplexes, G-quadruplexes, Z-DNA, and R loop) [6]. These events can ultimately lead to replication stress by stalling of the replication fork, followed by collapse or breakage [7,8]. At the origin of replication stress, multiple mechanisms are involved, including nucleotide pool imbalance, conflict between replication and transcription, and oncogene activation [7,8]. Nonetheless, these distinct systems can be connected. Moreover, DNA can be exposed to external insults such as ultraviolet light, ionizing radiation and a large variety of chemical compounds [9,10] that also include a wide class of chemotherapeutic drugs. In addition, lesions in the DNA can also arise from the activation of the immune system and the following inflammation signals which can additionally lead to the release of reactive oxygen species (ROS) altering biomolecules and genetic materials [11].

Furthermore, it is important to highlight that DNA is still the main pharmacological target for the therapy of many hematological and solid malignancies. Most anticancer drugs are either inhibitors of nucleic acid synthesis (e.g., antimetabolites) or inducers of DNA damage impairing DNA function (i.e., alkylating agents, platinum drugs, topoisomerase I or II inhibitors), thus their selectivity being related to a defective DDR. While the existence of a strong connection with the immune system is becoming more evident, the effect of chemotherapy upon the immunological responses is still poorly understood. 

Last but not least, the concept of “immune-therapy” has recently expanded based on the general knowledge that neoplastic cells express several neo-antigens that (at least in the initial phases of malignant transformation and tumour progression) constitute modified variants of the self that are not subject to central and peripheral tolerance [12]. Thus, either boosting the cancer immunogenicity and the ability of endogenous T cells to recognize neoantigens to destroy cancer cells either blocking the inhibitory signalling (immune checkpoint inhibitors, ICIs) that sedate T cells activation, both represent current strategies in cancer treatment employing the immune system. Interestingly, human malignancies with a high mutational load show a superior response to immunotherapy with checkpoint blockers than tumours with a relatively low number of somatic mutations, depending on the adaptive immunity response corroborated by the innate [12,13,14].

In this review we aim to mainly focus on mechanisms, either belonging to the classic DDR and to the innate immune system, that crosstalk and are activated when a DNA damage and aberrant DNA localization (in cytosol or endosomal compartments) occurs, indicating an infection or issues with nuclear integrity. We provide an overview of the agents that cause DNA damage, from solar radiation to DNA-interacting drugs whose effects are likely related to immunological mechanisms. Finally, we attempted to merge the complexity of different strategies currently employed in the clinic and that stand at the intersection between DDR and immunity.

## 2. DNA Damage and DNA Damage Response

DNA lesions such as nucleotide alterations (substitution, deletion, and insertion), bulky adducts, single-strand breaks (SSBs) and double-strand breaks (DSBs) can be induced by both endogenous and exogenous agents. The most common endogenous challenges are represented by transcription-replication conflicts (TRCs) due to the placement of transcription complexes within the replication forks, leading to their stalling, to DNA recombination, breaks and mutations [15,16,17]. For instance, the R-loop represents a typical transcription-association replication fork barrier, as nascent transcripts re-anneal to their template DNA and a RNA-DNA hybrid is then formed, displacing the non-template strand as single-stranded DNA (ssDNA) [18]. The loss of RNA processing and regulatory factors increases the R-loops, causing DNA damage and genome instability in eukaryotic cells [19,20,21,22]. 

DSB is a dangerous lesion that can be mutagenic due to chromosomal rearrangements or loss of genetic information as result of erroneous DNA repair. In response to DNA damage, a network of events becomes collectively activated to induce the DDR. It includes the DNA damage recognition through a set of sensing molecules, the activation of checkpoint proteins, and finally of repair systems that operate upon damage such as nucleases, helicases, polymerases, and ligases [23]. The molecular components of the induced DDR are typically classified into three major groups: “sensors”, “transducers” and “effectors”, which mediate eventual outcomes that cannot be taken for granted as repair, apoptosis and immune clearance. The DDR leading from DSBs activates a network of related pathways including Homologous Recombination (HR), Non-Homologous End Joining (NHEJ), Microhomology Mediated End Joining (MMEJ) and the Fanconi Anaemia (FA) repair complex or Interstrand crosslink repair (ICL) [24,25]. Beside DBSs, the most common type of lesion in cells are SSBs, which affect only one strand of the DNA and are usually associated with the loss of a single nucleotide [26]. In humans, the DDR has evolved to cope with SSBs through specific DNA damage response/repair pathways including Mismatch Repair (MMR) for mismatched bases, Base Excision Repair (BER) for base modifications, Single Strand Annealing (SSA) for single strand breaks and Nucleotide Excision Repair (NER) for intra-strand cross-links and thymidine dimers, which can be further classified as Global Genomic Repair (GG-NER) and Transcription-Coupled Repair (TC-NER) [24,27]. Finally, another simple but efficient system, named Direct Repair (DR), is able to reverse a damaged base to its native state by highly conserved specific enzymes [28,29]. All pathways are described in Table 1.

Overall, the DDR protects genome stability, coordinating a network of pathways ensuring the correct transmission of genetic material at any stage of DNA replication, repair and recombination, cell cycle checkpoint and chromosome segregation. DDR pathways do not play independently in the DNA repair machinery, but they rather cooperate, so that the impairment of one pathway is efficiently overtaken by the recruitment of other DDR networks. For this reason, when these pathways are perfectly functioning, the damage is successfully detected and accurately repaired, thus the cell is restored to normal functioning; otherwise, many outcomes are possible for DNA-damaged human cells. Under the circumstance of a persisted DNA damage, programmed cell death or apoptosis is activated eliminate cells with genome instability. Nonetheless, if the damaged DNA is mis-repaired, the cell can gain the function to evade senescence and death, resulting in a pool of cells bearing chromosomal aberrations or deleterious mutations with high oncogenic potential. The associated repair mechanisms obviously play a crucial role in carcinogenesis, since most oncogenic alterations in humans (mutations, translocations, amplifications, deletions, and epigenetic modifications) are caused by the inefficient repair of damaged DNA. In fact, DNA repair, DNA damage tolerance and DDR pathways are disrupted or deregulated in many cancers, thus the increases of mutagenesis and genomic instability is often addressed as the cause of cancer progression [19,20,43]. In the same way, aging is attributed to the wear and tear of chromosomal ends and failing capacities of a combination of these pathways to prove the errors [44,45]. Furthermore, neurodegenerative disorders also seem to be the result of a combinatorial failure of more than one of these processes [46,47].

DDR is not the only knight defending DNA integrity, but a concert of mechanisms is involved to ensure cell homeostasis, from the regulation of the redox balance to the activation of physiological pathways involved in the detoxification from toxic compounds or carcinogens. Additionally, it is fundamental to hold a delicate balance with different stimuli coming from inside as well as outside, where the microenvironment plays a crucial role in the activation of the immune system. The initial inflammatory response that is provoked by innate immunity can be beneficial or harmful depending on the type, strength and duration of the stimuli. While a weak activation results in the susceptibility to infections or tumor development, an excessive or self-antigen response can lead to autoimmune diseases or allergies [48,49].

## 3. At the Intersection of DNA Damage and Innate Immunity

Inflammation is a universal cell-intrinsic response to infections or tissue damage. For instance, when the innate immune system detects infection agents, it eliminates the initial cause of cell injury, and clears out necrotic cells and damaged tissues, and finally initiates tissue repair. The innate response is mainly mediated by the activation of phagocytic cells, such as non-professional immune cells (like epithelial cells, endothelial cells and fibroblasts) and professional APCs (like neutrophils, macrophages, and dendritic cells) [50]. At the very beginning, innate immunity was considered to be a nonspecific response, but researchers have shown that the innate response has a certain specificity, with the ability to discriminate between self and non-self molecules [50,51]. The initial step in immunity is indeed in the recognition of the foreign infectious agents by the host cells. To do so, sensor germline-encoded pattern recognition receptors (PRRs) are used to detect microbial products, such as microbial nucleic acids, lipoproteins, and carbohydrates, named pathogen-associated molecular patterns (PAMPs) or damage-associated molecular patterns (DAMPs) released from injured cells [52]. In the setting of microbial infection, PAMPs, highly conserved in diverse organisms but are absent in the host, activate PRRs that oligomerize and assemble large multi-subunit complexes that initiate signaling cascades. Innate DNA sensors can be functionally divided into DNA sensors mediating type I interferon (IFN) response and those mediating the inflammasome activation. 

Mammals have several distinct classes of PRRs including toll-like receptors (TLRs), retinoic acid-inducible gene-I (RIG-I)-like receptors (RLRs), Nod-like receptors (NLRs), absent in melanoma 2 (AIM2)-like receptors (ALRs), C-type lectin receptors (CLRs), and intracellular DNA sensors such as cyclic GMP-AMP synthase (cGAS) [48,53,54,55] and each distinguish a specific class of PAMPs that univocally lead to innate immunity activation, inflammation and recruitment of leukocytes to the region. 

Pioneering works have revealed the strong crosstalk between immune responses and DDR for the recognition of misplaced self-DNA. It has been proved that DNA damage can trigger innate immune responses through the accumulation of nuclear DNA in the cytoplasm [56,57]. Furthermore, a common feature of tumors and cancer cell lines is the accumulation of cytoplasmic ssDNA and double-stranded (dsDNA) DNA [57,58]. Many human tumors display chromosome instability (CIN) phenotype, a condition as a result of which chromosome missegregation happens at increased frequency, and often coincides with cytosolic DNA that activate the cGAS- sensor protein stimulator of IFN genes (STING) pathway, forming an essential node between cancer cells and the immune microenvironment. In fact, STING is considered a novel player with pleiotropic effects in the field of the immune system. Currently, the STING-targeted treatment is a candidate for anti-tumor immunotherapy and agents such as ADU-S100 (MIW815) (ClinicalTrials.gov, Identifier: NCT02675439), MK-1454 (NCT03010176), and E7766 (NCT04144140) have been approved for clinical trials to test their role in cancer progression in humans. Accordingly, an excitingly promising direction for cancer immunotherapy is the modulation of the STING pathway for increasing immune surveillance. This innovative strategy would necessarily act through the tumor microenvironment (TME) stimulation by IFN secretion and lymphocyte infiltration, which is reviewed elsewhere [59,60].

### Innate DNA Sensors and Signaling

The detection of aberrant nucleic acids has evolved as a crucial mechanism of host defense as nucleic acids are central to the replication of most pathogens. The accumulation of endogenous DNA by-products in the cytosol, such as those generated during DNA replication or derived from endogenous retroviruses [61] and the abnormal occurrence of bacterial or viral dsDNA in endosomes triggers immune activation [62,63]. An intriguing question is how sequence-independent DNA sensors are able to distinguish between foreign and host DNA, given that nucleic acids are not only unique to pathogens. Mammalian cells overcome this issue by recognizing differences in the physicochemical structure of microbial DNA and self-DNA, such as the high frequency of immunostimulatory CpG motifs that can be easily spotted in bacterial DNA, as well as the secondary structure of AT-rich DNA in Plasmodium falciparum genome, both critical to stimulate immunity [64,65,66]. In detail, human TLR9 is the only known receptor expressed in plasmacytoid dendritic cells and B cells (whereas mouse TLR9 is expressed in plasmacytoid, myeloid, B cells and macrophage) that recognizes bacterial DNA rich in unmethylated CpG motifs [48,52,53,64]. Before activation, the receptor is found in the endoplasmic reticulum but on exposure to CpG DNA, TLR9 rapidly translocates to endosomes. Subsequently, it transmits the signal through the TLR adaptor myeloid differentiation primary response gene 88 (MyD88), which, in turn, activates the transcription factors nuclear factor kappa B (NF-kB) and IFN-regulatory factor (IRF) 7. Those are key factors for the induction of inflammatory genes such as the tumor necrosis factor (TNF), interleukin (IL)-1 or IL-6, and type I IFNs [67]. In the absence of TLR9, other mechanisms are capable to trigger the immune response [68]. Indeed, nucleic acids can also be sensed in other cell compartments such as endosomes and detected by other specific TLRs like TLR3, TLR7, TLR8 and TLR13 leading to the same type I IFN gene transcription mediated by the same or different adaptor and effector proteins [54,69,70]. The first discovered cytoplasmic sensor was the DNA-dependent activator of IFN regulatory factor (DAI) that binds synthetic dsDNA and engages the TANK-binding kinase 1 (TBK1)–IRF3 cytosolic DNA-sensing pathway to regulate the type I IFN response [71,72]. Earlier, the expression of DAI was shown to be greatly up-regulated in the peritoneal lining tissue of tumor-bearing mice due to INF-γ or lipopolysaccharide (LPS) activation in macrophages, suggesting that this protein plays a role in host defense [73]. Cytosolic DNA can also be detected by the interferon gamma-inducible protein 16 (IFI16). This latter functions as a DNA sensor in both the nucleus and the cytoplasm [74], and as a nuclear pathogen sensor upon infection with Kaposi Sarcoma-associated herpesvirus, which maintains its latency and repression of lytic transcripts via IF116 [75,76]. However, the AIM2 protein detects cytosolic DNA activating the inflammasome pathway, instead of type I IFN gene transcription as the previous, promoting pyroptosis and the caspase 1-mediated maturation of inflammatory cytokines such as interleukin-1β (IL-1β) and IL-18 [77]. Additionally, aberrant cytosolic RNA activates the family of RIG-I-like receptors that recognizes double-stranded RNAs (dsRNAs) that have a 5′-triphosphate and are either viral in origin or generated by RNA polymerase III from microbial DNA templates and that lead to type I interferons production along with other pro-inflammatory cytokines [53]. Furthermore, RNA polymerase III, which is typically known to transcribe small RNAs, including 5S rRNA and tRNAs, also functions as a DNA sensor through its binding to AT-rich dsDNA in the cytosol [78]. Once it engages its ligands, it transduces the signal to the nucleus for the production of proinflammatory cytokines. 

A major cytosolic ssDNA and dsDNA sensor is known as cGAS-STING. The receptor protein GMP-AMP synthase is able to sense cytosolic dsDNA, and as response, synthesizes secondary messenger 2′,3′-cyclic GMP-AMP (cGAMP). This latter is detected by the downstream sensor protein STING, triggering IRF3 activation for the production of type I IFN. The NF-κB signaling can also be activated by STING. In detail, the activation of cGAS is regulated by dsDNA in a length-dependent manner, since long DNA fragments (kilobase range) activate more efficiently cGAS than shorter ones. Moreover, cGAS also guides the downstream production of IFN with longer portion of DNA being more immunostimulatory [79]. Several excellent reviews have speculated about the overlapping roles of cyclic GMP-AMP synthase (cGAS)-STING between pathways of the innate immunity as well as of the DNA damage repair [80,81]. 

Another factor that recognizes cytosolic self-DNA and viral DNA is DExD/H-box helicase 41 (DDX41). Originally, it was found in the cytosol of myeloid dendritic cells (mDCs), together with STING. When it was silenced by shRNA, mDCs failed to mount type I interferon and cytokine responses, and moreover, the subsequent TBK1, NF-kB and IRF3 activation were completely inhibited [82]. Considering it all together, the activation of various cytosolic sensors is essential to guarantee a proper response to misplaced or damaged DNA. The detection of nucleic acids, if not controlled but exacerbated, can be harmful and dangerous for the host, leading to tissue damage, energy consumption, and the promotion of autoimmune diseases. Thus, fast and effective mechanisms for degrading DNA have evolved in parallel, such as DNases. Specifically, DNase II is known to be a potent lysosomal defense barrier that rapidly degrades DNA derived from pathogens or apoptotic cells. In fact, it is used under homeostatic conditions or when the apoptotic enzyme caspase-activated DNase (CAD) fails to digest the chromosomal DNA of apoptotic cells engulfed by macrophages and the cells need to quickly degrade DNA molecules avoiding immune sensors activation. Another DNase, three prime repair exonuclease (TREX) 1, was discovered to be a function primarily in the replication and DNA repair, degrading DNA products derived from endogenous retroviruses or DNA replication [61,83]. Defects in TREX1 leads to the accumulation of foreign and self-DNA products that are thought to be the cause of autoimmune disorders [84]; indeed, for example, a loss of function mutation in the human Trex1 gene causes Aicardi–Goutieres syndrome (AGS) [85]. Most of the cited sensors have been described in Figure 1.

Any event that could potentially activate DDR can stimulate the innate immune pathways, even in the absence of DNA damage; thus, when tolerable DNA damage levels occur, the modulation or suppression of DDR signaling can mitigate some pathological consequences of DNA damage-driven inflammation.

Nuclear DNA damage is recognized by a set of several sensors, including: RPA (Replication Protein A) detecting single-strand breaks, the MRN complex (Mre11-Rad50-Nbs1) sensing double-strand breaks, and the MutS proteins recognizing mismatched bases. Additionally, DNA-PK (DNA-dependent protein kinase) and Ku70/80 are able to bind DNA to trigger the DDR, depending on the lesion; nonetheless, they can also be found in the cytosol activating the IFN response, mediated by IRFs (IFN-regulatory factors). Other DDR proteins such as ATM (ataxia telangiectasia mutated), once phosphorylated, shuttle to the cytoplasm and activate the NF-kB (nuclear factor kappa B) mediated response. The nuclear complex ATR (ATM- and Rad3-Related), together with ATR interacting proteins (ATRIP), are also ready to respond to a broad spectrum of DNA damage, including DSBs and a variety of DNA lesions that interfere with replication. Additionally, the protein Rad50, in a complex with CARD9 (Caspase Recruitment Domain Family Member 9), promote the NF-kB response. Likewise, RNA pol III acting like a sensor induces the same. In the endosomes, the presence of dsRNA, ssRNA, and CpG DNA stimulate and activate, respectively, the toll-like receptors: TLR3, TLR7/8, and TLR9. TLR9 and TLR7/8 recruit the MyD88 (myeloid differentiation marker 88) inducing the transcription of NF-kB, while also triggering the IFN response via IRF7. TLR3 instead activate STING (cGAS- sensor protein stimulator of IFN genes) via TBK1 (TANK-binding kinase 1), promoting the IFN response. Many are also the sensors for dsDNA in the cytoplasm. AIM2 (absent in melanoma 2) binds dsDNA and activates the inflammasome for the release of interleukin (IL)-1β. DAI (DNA-dependent activator of IFN regulatory factor), cGMP- dependent cGAS (Cyclic GMP-AMP synthase), DDX41 (DExD/H-box helicase 41), or IFI16 (Interferon gamma-inducible protein 16) may bind the cytosolic DNA and activate the type I IFN response, through TBK1/STING, as well for the MRE11/Rad50 complex and DNA-PK. This figure was created with BioRender.

## 4. DNA Damage Inducers

### 4.1. From Carcinogens to Radiation and Chemotherapy

Endogenous processes, as well as environmental and/or lifestyle factors, can provoke DNA damage, thus leading a pivotal role in cancer etiology. Many of the wide range of DNA-damaging agents, to which we are continuously exposed, are classified as carcinogens and are associated with various forms of DNA damage such as SSB, DSB, covalently bound chemical DNA adducts, oxidative-induced lesions and DNA–DNA or DNA–protein cross-links.

Carcinogens are categorized as chemical or physical agents causing DNA damage attributable to their properties. They span from the polycyclic aromatic hydrocarbons, heterocyclic aromatic amines, to mycotoxins, to the ultraviolet radiation (UVR), ionizing radiation (IR), aristolochic acid, nitrosamines, asbestos and nanoparticles [86,87,88,89]. 

Generally, chemical agents directly interface with DNA and other cellular components through their electrophilic groups, which allow for the interaction with negatively charged cellular macromolecules, thus leading to molecular alterations [87,90,91,92]. However, the physical presence of silicate minerals (named asbestos), which are composed of long and thin fibrous crystals, were found to be highly carcinogenic; indeed, the continuous use in industry and household applications was directly linked to asbestosis, pleural plaques and mesothelioma. The related damage occurs through oxidative stress leading to DNA strand breaks, fibrosis and interaction with the mitotic apparatus of dividing cells and a low grade of chronic inflammation [93,94,95].

Carcinogens like aristolochic acid (AA), heterocyclic aromatic amines (HAAs), polycyclic aromatic hydrocarbons (PAHs), N-nitrosamines, and mycotoxins are relatively unreactive compounds, since they require bioactivation to exert genotoxic effects [96,97,98]. While these indirect carcinogens are reliant on activation, a few can enhance bioactivation through the modulation of gene expression. For example, PAHs increase the expression of CYP450 family by acting as exogenous ligands of the cytosolic aryl hydrocarbon receptor (AhR)–aromatic receptor nuclear translocator complex, which mediates toxic responses to dioxin-like environmental contaminants, and nonetheless, regulates of the development and function of both innate and adaptive immune cells [96,99,100]. Furthermore, AhR is a transcription factor that, once ligand-activated, translocates to the nucleus, and regulates the expression of target genes, depending on the cell type, expression level, tissue microenvironment or concurrent events such as inflammation. Notably, it is abundant in all skin cells and is highly expressed in T helper 17 (Th17) cells, modulating their expression of IL-22; additionally, it is crucial for the balance between inflammatory and regulatory T (Treg) cells [101,102]. In epidermal cells, UVBR exposure can activate AhR signaling in response to the formation of tryptophan photoproducts, in particular 6-formylindolo[3,2-*b*]carbazole (FICZ), which binds with high affinity to the receptor and activate downstream signaling pathways. AhR antagonists and *Ahr*- knockout mice are resistant to UVR- induced local immunosuppression, which indicates that AhR is essential for some of the UVR- mediated immunosuppressive mechanisms, such as the production of cytokines, changes to the induction of Treg cells or antigen- presenting cells (APCs) basal activity [103,104].

Moreover, physical agents such as UVR and IR are considered to be damaging agents. UVR has traditionally been viewed as harmful due to its mutagenic properties and is dangerous to proteins and lipids chemistry, thus promoting carcinogenesis and contributing to aging. The sun, which emits radiation in the UVA (320–400 nm), UVB (280–320 nm) and UVC (<280 nm) wavelengths, is the primary source for UVR. Overall, UVR is important for normal physiology, mediating melanogenesis, vitamin D production, cell growth and differentiation; however, at large doses, UVR can cause sunburn and hyperplasia, and chronic exposure can cause skin aging and increases the risk for melanoma and basal cell carcinoma [10].

Molecular and cellular damage pathways, which are induced when UVR is absorbed by chromophores present in the skin, cause damage in target cells and tissues. Indeed, UVR induces the formation and release of DAMPs from necrotic keratinocytes that activate TLR signaling, which leads to the inflammatory response in neighboring healthy keratinocytes and the production of antimicrobial peptides (AMPs) as response, indicating that a sort of insult has occurred nearby [10,105]. As result of the UVR-induced epidermal damage, keratinocytes produce and release immunomodulators, including *cis*- urocanic acid (*cis*- UCA), platelet-activating factors (PAFs) and PAF- like ligands, IL-10, FICZ, epidermal- derived receptor activator of nuclear factor-κB ligand (RANKL) and TNF [10,106,107]. Consequently, more soluble mediators are produced and multiple signaling pathways and cells get activated.

Collectively, the signaling pathways induced by UVR exposure lead to an immunosuppressive environment, and this property is at least partially responsible for skin carcinogenesis in both animal models and humans [108]. The immunosuppressive phenotype is featured by high levels of TNF, IL-4 and IL-10 and is associated with Langerhans cell migration to the lymph node and neutrophil recruitment to the skin [109,110], the induction of a T helper 2 (Th2) cell response [111], as well as IL-4-producing natural killer T (NKT) cells, Treg cells and regulatory B (B reg) cells. These latter secrete IL-10 and suppress dendritic cells (DCs) function, promoting cell- mediated and humoral immunosuppression [112,113,114].

The capability to alter or inducing DNA damage has been exploited in the last decades and is currently used in cancer treatments. Indeed, together with surgery, chemotherapy and radiation represent the major treatments in oncology. The reason lays in a simple rationale: proliferating cancer cells are usually more sensitive to chemotherapy or radiation than normal cells as they gain DNA mutations and defects in DDR mechanisms, thus being unable to correct the errors and proceed in the cell cycle. If a high level of DNA damage occurs, cell-cycle checkpoint proteins become activated and arrest the cell-cycle, thus preventing the transmission of damaged DNA during mitosis. When DNA lesions occur during the S phase of the cell cycle, replication fork progression is blocked because it can be responsible for the replication-associated DNA DSBs, which are among the most toxic of all DNA lesions. If the damaged DNA cannot be properly repaired, cell death may result [115]. Historically, the development of numerous anticancer compounds, such as cisplatin, methotrexate, doxorubicin, 5-fluorouracil, etoposide, anthracyclines, and gemcitabine have followed the concept of aiming at DNA as a target for anticancer drugs, directly or indirectly inducing DNA modifications [116].

In line with chemotherapy, the use of high-dose IR is currently one of the most common modalities in the treatment of many types of cancer. High precision techniques have made treatment delivery more effective and safer for adjacent normal tissue. Human exposure to IR is classified according to the doses, below 0.1Gy are classified as “low”, while doses normally used in medical procedures, such as Radiation Therapy (RT) (2-3Gy), are classified as “high”. Overall, photon beams in the form of low Linear Energy Transfer (LET) radiation (X-rays, Gamma-rays) are the main therapeutic modality employed in RT, although high LET radiation (protons, alpha particles, and other heavy ions) are sometimes considered thanks to their precise dose localization. Obviously, radiation-induced cell death is modulated by several factors, such as dose and irradiation schedules, which can trigger the formation of free radicals damaging the DNA, in turn promoting cellular stress with the final outcome of cellular senescence or death. While the effect of radiotherapy on the immune system has not been largely studied, such as chemotherapy, luckily there is growing interest in research about the anticancer immune response activated by ionizing radiation that is trying to fill the gap. In fact, it seems reasonable to explore a combination of different treatment strategies in this promising field [117,118].

### 4.2. DDR Induced Oxidative Stress: Meaning for the Host Immune Response

In homeostatic conditions, ROS are physiologically produced and act as signaling molecules within the cells. To avoid toxic oxidative stress, the balance of oxidants and antioxidants must be in place. For this purpose, healthy cells use enzymatic or non-enzymatic antioxidants such as tocopherols, vitamin E, glutathione (GSH), ascorbic acid (AA) and vitamin C (vit C) to compensate for the activity of oxidative agents. Among them all, the AA and the reduced form of GSH play the primary role in fighting against ROS as well as maintaining a normal oxidative balance.

In chemistry, the reaction of water radiolysis involves the formation of intermediates, partially reduced oxygen species, which give rise to hydroxyl radicals (OH·), singlet oxygen or reactive nitrogen species, which are collectively termed ROS. They can be produced endogenously by several organelles such as mitochondria (where O_2_ acts as a terminal electron acceptor for electron transport chain), peroxisomes (which contain enzymes that produce H_2_O_2_ e.g., polyamine oxidase), endoplasmic reticulum (produce H_2_O_2_ as a byproduct during protein folding) and by the cell membrane bound enzyme nicotinamide adenine dinucleotide phosphate (NADPH) oxidase.

While they show physiologic important cellular roles, they can trigger a number of adverse biological reactions by attacking structural and functional molecules, causing defects in DNA synthesis and repair mechanisms, as well as inactivating various key proteins and repair enzymes. In detail, hydroxyl radicals can indirectly produce SSBs and modify both the bases and sugars in the DNA molecules, as well as crosslinks between two complementary DNA strands, which can be cytotoxic or mutagenic. Indeed, the addition of OH· at position C8 within the guanine ring generates the oxidative product 8-oxo-7,8-dihydro-20-deoxyguanosine (8-oxodG), and the addition of OH· at position C8 of deoxyadenosine generates the oxidative product 8-oxo-7,8-dihydro-20-deoxyadenosine (8-oxodA), both capable of further oxidation forming 8-oxoG and 8-oxoA, non-coding mutagenic DNA bases [119]. Other oxidative products are thymine glycol and cytosine glycol, which, upon deamination, also lead to the formation of uracil glycol; thymine glycol induces conformational alterations that modify telomeres, with 8-oxodG also playing an additional role [120]. These modified bases are removed by DNA glycosylase enzyme through BER but their accumulation over time enhances genomic structure defects and instability. Chemotherapeutics such as alkylating agents (e.g., cisplatin, cyclophosphamide, and trabectedin) or antibiotics (e.g., doxorubicin) or antimetabolites (e.g., methotrexate and 5-fluoroacil) exploit this effect upon cancer cells, increasing ROS levels that contributes to their genotoxicity [121].

Within the cell, stress-induced ROS operates as a powerful alarm signal for the activation of defense mechanisms, where secondary messengers modulate the activation of the NFE2-related factor 2 (NRF2)-mediated antioxidant response. A versatile antioxidant system rigorously controls ROS activity modulating their intracellular concentration; although, when stress is prolonged, ROS concentrations overcome the scavenging action of the antioxidant system, resulting in extensive cellular damage and necrosis [122].

Additionally, the extent of DNA damage is not limited to the nucleus but easily reaches the mitochondrial DNA (mtDNA) that is more susceptible to damage than the nuclear because of the lack of repair proteins and other higher order structures, and because of the proximity to sites that generate reactive species. Their accumulation induces mitochondrial DNA lesions, strand breaks and degradation of mitochondrial DNA; in fact, ROS represents an important cause of the mtDNA mutations, which accumulate with aging and in diseased states. It also seems that the inhibition of BER enhances mtDNA degradation in response to both oxidative and alkylating damage [123,124].

Rather than the hazardous byproduct of mitochondrial respiration, the functional roles for ROS in cells has been elucidated as acting as signaling molecules (e.g., H_2_O_2_ regulating NFκB, MAPK pathways) to aiding immunity (e.g., oxidative bursts in phagocytes to eliminate pathogens) in References [125,126]. Mitochondria-derived ROS and released mtDNA directly induces the activation of innate immune responses such as sGAS-STING, and NF-kB signaling pathways. Injury-induced stress molecules ROS, ATP, mtDNA, and harmful environmental substances such as silica and asbestos, K^+^ efflux, and lysosomal destabilization, as well as DNA or RNA, are inducers of the NLRP3 (NOD-, LRR- and pyrin domain-containing protein 3) inflammasome (NLR family, pyrin domain-containing 3), a large multimolecular complex that exerts cytosolic surveillance and intensifies the inflammation.

Activation of the inflammasome NLRP3 requires two signals: (a) priming with either TLR or NLR ligands to enhance NF-κB-driven transcription of NLRP3, and (b) the exposure to microbial toxins, ionophores or endogenous alarmins to trigger the inflammasome assembly [122]. When activated, the inflammasome modulates the proteolytic enzyme caspase-1, resulting in the maturation of proinflammatory cytokines, such as IL-1β and IL-18, and their secretion [127] (as in Figure 1). It is believed that the NLRP3 inflammasome is able to recognize cytosolic nucleic acids and other endogenous danger signals indirectly. Indeed, it has been suggested that NLRP3 activation might also be triggered by perturbed cell membranes [128]. Given the diverse types of stimuli converging to NLRP3, it has been hypothesized that ROS might be the direct mediator that triggers NLRP3 activation [129]. It has also been confirmed when the inhibition of NADPH oxidase, induced by ROS, inhibited NLRP3 activation in macrophages treated with ATP [130]. Further, the absence of the p22*phox* subunit within NADPH oxidase substantially attenuated IL-1β production when macrophages are exposed to asbestos [131].

In addition to immune cells, mitochondrial dysfunctions can also extensively influence the function of non-immune cells. Indeed, as a result of mitochondrial dysfunction, endothelial cells secrete multiple pro-inflammatory cytokines such as IL-1, IL-6, and tumor necrosis factor-alpha (TNF-α), and upregulate intercellular adhesion molecule-1 (ICAM-1) expression, which attracts monocyte activation and adhesion, leading to a sterile inflammation called senescence associated secretory phenotype (SASP) [132,133].

## 5. DNA Damage and Inflammation: A Strong Interplay

Inflammation is the first reaction of tissues in response to harmful stimuli, such as pathogens, damaged cells, or other stressors. Inflammation induces the alarm, production of several factors, that in synch and in synergy modulate blood vessels permeability, recruit leukocytes, and create the context for the activation of innate first and adaptive immune responses after. The initial phase is usually named acute and it can be seen as a protective process that normally results in the removal of the initial damaging cause, pathogens and dead cells, and finally leading to its resolution and tissue healing. In contrast, when pathogens or stress factors are not removed, the resolution phase of inflammation does not occur and unnecessary tissue damage further fuels inflammatory processes. Indeed, chronic inflammation is thought to generate an excess of ROS and nitrogen species (RNS) triggering DNA damage and diseases. To date, chronic inflammation is associated with the onset and/or worsening of several diseases, including cancer, arthritis, colitis, diabetes, atherosclerosis, age-related degeneration and neurodegenerative diseases. Chronic inflammation, that can either be induced by environmental agents or due to autoimmune/inflammatory diseases and chronic infections, is involved in both cancer development and progression, as demonstrated by experimental and clinical studies [134,135,136]. As proof of the tight interplay between DDR and immunity, the evidence has shown that a prolonged or elevated inflammation can be the result of a dysregulated DDR or of an accumulation of unrepaired DNA damages. For example, cell lines from patients with systemic lupus erythematosus (SLE) have a defective DSB repair, and thus the DNA damage promotes the autoimmune disease [137]. Furthermore, persistent DNA damage signaling in murine models carrying a defective NER, only in the adipose tissue, has been shown to trigger a chronic autoinflammatory response followed by fat depletion and metabolic abnormality [138].

The DDR leads to different types of fate such as apoptosis, transient cell cycle arrest or cellular senescence. Transient cell cycle arrest has a protective effect against tumorigenesis as it allows cells to accurately repair DNA damage before cell cycle progression; instead, persistent senescence can induce a secretory phenotype that can be ambivalent in inducing the proliferation of tumor cells and promoting the activation of the innate immunity. The activated innate immune system, in turn, can suppress tumorigenesis by clearing senescent cells with oncogene activation or chronic DNA damage via the production of ROS and RNS that, if persistent in the microenvironment, can promote chronic inflammation. This latter has recently emerged as an important modulator of mutation susceptibility and is also known to drive aging and age-associated pathologies. The new term “inflammaging” was indeed conceived to better describe a chronic, low-grade inflammation, carrying a highly significant risk factor for both morbidity and mortality in the elderly people, given that most age-related diseases share an inflammatory pathogenesis [135,139].

As a matter of fact, DNA damage driven inflammation can also promote tumorigenesis. The connection between inflammation and cancer, was first perceived in the nineteenth century and is now accepted as an enabling characteristic of cancer [134]. Indeed, it is generally known that up to 25% of human malignancies are related to chronic inflammation and to viral and bacterial infections. Cancer-related inflammation represents the seventh hallmark in the development of cancer as the well-described cancer-related chronic inflammation fosters unlimited replicative potential, independence of growth factors, resistance to growth inhibition, escape of programmed cell death, enhanced angiogenesis, tumor extravasation, and metastasis [140]. The connection between chronic inflammation and tumorigenesis is further supported by findings that mark the inflammatory mediators as a cause of genetic instability in cancer cells. Nonetheless, non-resolving inflammation is one of the consistent features of the tumor microenvironment; in fact, tumors that are not epidemiologically related to inflammation at least are also characterized by the presence of inflammatory cells and mediators [134,141,142]. Examples of key players of cancer-related inflammation (CRI) include tumor-infiltrating lymphocytes, tumor-associated macrophages (TAMs), the secretion of cytokines such as TNF, IL-1, IL-6, and chemokines, such as CCL2 (C-C Motif Chemokine Ligand 2) and CXCL8 (C-X-C Motif Chemokine Ligand 8), in addition to the occurrence of tissue remodeling and angiogenesis [140]. The secretion of cytokines activates the oncogenic transcription factor NF-kB and Signal Transducer And Activator Of Transcription 3 (STAT3), both inducing the expression of target genes crucial for tumorigenesis such as anti-apoptotic genes, stress-response genes and pro-angiogenic molecules [141]. Usually, chronic inflammation is followed by the generation of ROS/RNS that, in turn, provoke oxidative DNA damage and the impairment of DNA repair pathways, thus promoting cell transformation. For instance, patients with Crohn’s disease (CD) and chronic intestinal inflammation were reported to accumulate high levels of ROS/RSN in colonic mucosa, correlated with disease severity and colorectal cancer [143,144]. In asbestosis, inflammatory condition triggered by the exposure to asbestos, neutrophils and macrophages release ROS, increasing the level of oxidative DNA damage and causing malignant mesothelioma (MM) in tumor arising from mesothelial cells [93,94,95,145].

## 6. Targeting DDR to Defeat Cancer

Frequently, cancer cells show high levels of DNA damage, loss of one or more DDR pathway and increased DNA replication stress. These features can lead to specific DDR vulnerabilities that can be exploited as potential therapeutic targets using different approaches, such as blocking DDR and maximizing the DNA damage in cancer cells, exploiting DDR dependencies, and targeting DDR proteins associated with replication stress response. Thus, targeting the DDR machinery is an attractive strategy for designing novel chemotherapeutics and combinatorial approaches.

The efficacy of this strategy mainly relies on the concept of synthetic lethality, which refers to the co-occurring gene mutations leading to cell death. This applies to DDR when an impaired DNA-repairing gene causes the recruitment of other DNA damage networks. Thus far, HR has been the most targeted pathway. Among its main actors, Breast Related Cancer Antigen 1/2 (BRCA1/2) play a key role in fixing double strand defects by an error-free mechanism, thus functional impairment due to mutations in these genes lead to the activation of alternative pathways with the consequent accumulation of many genomic aberrations. BRCA1/2 are mainly mutated in gynecological cancers like ovarian and breast, however they have also been reported in peritoneal, prostate and pancreatic cancer [146,147]. Tumor-bearing BRCA defects have shown high sensitivity to platinum-based drugs [147,148] and to other DNA-interacting agents like trabectedin [149], and they have been shown to guide the choice of using platinum-based therapy in non-small cell lung cancer [146]. Moreover, a defective function associated to these genes resorts to synthetic lethality through the molecular mechanism associated to Poly-(ADP-ribose)-polymerase 1 (PARP1). In fact, when PARP1 is inhibited and trapped in the DNA, base excision repair is inhibited with the consequent formation of unrepaired ssDNA breaks that can lead to replication fork stalling and the formation of the dsDNA breaks usually repaired with HR. Cancer cells that have deficiencies in HR, for example due to mutations in the tumor suppressor genes BRCA1 or BRCA2, are unable to repair DNA damage when PARP1 is inhibited. This results in the accumulation of errors, genomic instability and the final death of the cancerous cells [150]. The strong interconnection between BRCA1/2 and PARP1 justifies the use of PARP inhibitors (PARPi) in BRCA-defective tumors. PARPi like olaparib, niraparib and rucaparib have received FDA approval for the treatment of ovarian and breast cancers [146], however they have also been tested in gastrointestinal cancers carrying defects in BRCA1/2 or ATM [150]. Recent studies also suggest high responses in *non*-BRCA mutated cells. This could be explained by mutations in other genes belonging to HR, like ATM, ATR, RAD51, BARD1 [150], and by genomic instability like tumor allelic imbalance (TAI) and loss of heterozygosity (LOH), indicative of homologous recombination deficiency (HRD) [151,152,153]. A recent study by Mirza et al., analyzed four phase III clinical trials testing PARPi as monotherapy or in combination, evaluating both BRCA and HRD-tumors [151]. Although with heterogeneous results, all studies suggest the importance of the evaluation of the HRD status in addition to BRCA1/2 for PARPi treatment [151]. The role of *non*-BRCA genes mutations and genomic instability in PARPi response is also advised for a pharmacological application of these inhibitors in other cancer types. A recent review investigated the role of DNA damage in malignant pleural mesothelioma (MPM) [154] where MPM seems to exhibit mutations in DDR at both the germline and somatic levels with frequent mutations in BRCA1 associated protein 1 (BAP1). Despite the lack of solid evidence of the efficacy of PARPi in MPM, Fuso Nerini et al. suggest that further investigation on the response to PARPi should be considered as preliminary results with rucaparib in a selective cohort of patients with BRCA1/BAP1 inactivating mutations seem to be promising [154]. Another application of PARPi to *non*-BRCA tumors concerns sarcomas: a clinical trial in phase Ib where olaparib was administered in combination with trabectedin showed activity regardless of *BRCA* status, thus widening the possibility of exploiting PARP1 potential (NCT02398058) [155].

Two well-known damage sensors that reside at the basis of HR are ATM and ATR. While activated by different type of damage, ATM by DSBs and ATR by SSBs, their role in DDR is strictly interconnected, and thus they represent a prime target for inhibition. Pilié et al. revised novel inhibitors of ATR like M6620, M4344 and AZD6738 and showed the high toxicity of these drugs when used in combination [150]. However, M6620 and AZD6738 were also considered in a recent review in which Mei et al. concluded that the use of these inhibitors could serve as a rescue therapy for patients who have experienced tumor progression with PARPi [156]. The role ATM-inhibitors is otherwise enhanced when used in combination with PARPi [157]. Indeed, despite the efficacy shown in vivo and in vitro studies, a recent study associated to a phase I clinical trial (NCT02588105) assessed that the use of ATM inhibitors as monotherapy had low antitumor effects [150], while pre-clinical studies in cell lines showed that the combination of the novel ATM inhibitor AZD0156 in combination with PARPi leads to an increase in DNA double-strand break signaling, cell-cycle arrest, and apoptosis [158]. Besides HR, DBSs are fixed by the NHEJ repair machinery, in particular those caused by radiation therapy [150]. As it is known, DNA-PKcs is a key enzyme in NHEJ and mutations in this protein causes high sensitivity to ionizing radiation. First generation DNA-PKcs have exhibited high toxicity, thus not reaching final clinical evaluation [159]. Selective and more promising inhibitors like M9831, medisertib and CG115 are under study as monotherapy or in combination with doxorubicin and IR [150,159]. The DNA repair and the cell cycle pathways are strictly intertwined. The repair machinery is able to fix DNA damages if cells have been prevented from going through their cycle progression. Thus, inhibition of cell cycle checkpoints like CHK1, CHK2 and WEE1 has been extensively studied in recent years. The main issue with inhibitors of CHK1/2 is related to low selectivity and toxicity, as described for UCN-01 and AZD7762 [150]. Additionally, a novel inhibitor named CCT245737 (SRA737) has been tested in non-small cell lung cancer and colorectal cell lines showing high sensitivity in combination with B-family polymerase inhibitors [160]. This drug is currently under clinical testing as monotherapy or in combination with gemcitabine (NCT02797977) [150]. However, the inhibition of WEE1 has captured more attention, and in fact, in the first half of 2020, more than 50 studies have been published investigating the role of WEE1 inhibitors in cancer. WEE1 inhibits CDK1/2 and controls the activation of G2/M cell cycle checkpoint. Indeed, adanosertib (AZD1775) is currently under study in clinical trials, both as monotherapy before surgery in high-grade ovarian, fallopian tube, and in primary peritoneal cancer and SETD2-deficient tumors (NCT03284385).

In addition, molecular alterations in genes involved in DNA mismatch repair (MMR) promote cancer initiation and foster tumour progression [43]; nevertheless, cancers deficient in MMR frequently show favourable prognoses and indolent progression [161]. Indeed, when the MMR machinery is defective, cancer cells display characteristic microsatellite instability (MSI) [162] that increases the tumour mutational burden (TMB) and favours the response to ICIs therapy, in the case of melanoma, bladder and lung cancers [163,164], but not in other hyper-mutated tumors as non–small cell lung cancer [165]. In general, increasing the number of mutations and neoantigens per se might not be sufficient to trigger tumour detection and immune surveillance as, in principle, cancer cells should accumulate an enormously high number of de novo mutations to generate such a load of neoantigens. Surprisingly, Germano et al. showed that it is possible to inactivate DNA repair in vivo to improve immune surveillance and response to immune-checkpoint blockade, when restricted to a clonal population. Accordingly, inactivation of MMR causes a hyper-mutation condition that increases tumour neoantigens, which trigger long-lasting immune detection enhanced by immune modulators [166].

Innovations in immunotherapies have revolutionized the concept of cancer treatment. Cancer immunotherapy is generally classified into two main classes, including active and passive approaches. Active interventions improve the immune system response of the patient (for example, vaccination or adjuvant therapy) promoting antitumor effector mechanisms with cancer elimination and immune cell death (ICD) [12,13,167]. Passive interventions involve the administration of specific monoclonal antibodies (mAbs) against different immune checkpoints as well as the adoptive transfer of genetically modified specific T cells called Chimeric Antigen Receptor T cell therapies (CAR-T). The latter are currently the most rapidly developing approaches for cancer targeted therapy [168,169]. Indeed, the mechanism of action of ICIs like anti-cytotoxic T lymphocyte antigen 4 (CTLA4) and anti-programmed cell death protein 1/programmed cell death ligand 1 (PD-1/PD-L1) has been widely explored. Recent works showed that PARP inhibition, in combination with ICIs, such anti-PD-1/PD-L1, could be effective for BRCA1-deficient tumors by activating antigen presenting cells such as dendritic cells via the cGAS-STING pathway [170,171,172]. Mechanistically, defining PARP-1 dependent DNA processing functions is pivotal for the development of successful PARPi therapy and therapeutic regimens in the treatment of several different cancers.

### PARPs at the Intersection of DNA Damage and Immunity

PARPs have been studied as potential targets for drug development, with PARP1, and more recently PARP14, attracting the most attention in the cancer field [173,174].

PARP-1 is a nuclear chromatin-associated protein functioning as DNA damage–sensor. It belongs to evolutionary-conserved family of proteins named Poly(ADP-ribose)-polymerases (PARPs), or more recently, ADP-ribosyl-transferase diphtheria toxin-like (ARTDs), involved in DNA repair, abiotic and biotic stress responses, cell death, division, and differentiation, as well as in inflammation and immune responses. [175,176]. It becomes catalytically activated when encountering free DNA ends, and starts to generate high amounts of PARs, which function as the scaffold for the recruitment of DNA repair enzymes. Additionally, self-modifications are used to induce its own dissociation from the site of DNA damage and localization of repair proteins to the lesion.

Genetic studies have originally described PARP-1 as part of the BER pathway, but more recent results unveiled a wider role for PARP-1 and PARylation in all major DNA repair mechanisms (e.g., NER, cNHEJ, aNHEJ, HR, MMR, and MMJ) and maintenance of replication fork stability [177,178,179,180,181]. By means of poly-ADP-ribosylation (PARylation), PARP-1 negatively regulates its own enzymatic activity [182]. Nonetheless, PARP-1 (as other family members) is also sensitive to other kinds of post-transcriptional modifications and can also be activated in response to signals independently of DNA damage. For instance, the extracellular signal-activated kinase (Erk) phosphorylates PARP-1 modulating its activity and consequently affecting NF-kB PARylation levels [183]. Nonetheless, PARP-1 is involved in NF-kB activation triggering IKKγ (NEMO, NF-kB regulator) SUMOylation and mono-ubiquitination, which allow for IKK and NF-kB activation [184,185].

Interestingly, PARPs are also involved in inflammatory processes, such as PARP inhibitors, and display protective effects in non-oncological diseases such as acute and chronic inflammatory diseases [173]. In effect, it is involved in modulating gene expression and activation of innate (neutrophils, macrophages, dendritic cells, and microglia) and adaptive (T and B lymphocytes) immune cells.

In detail, PARP-1 sustains the expression of pro- inflammatory mediators such as TNF-α, IL-1, IL-6, interferon-γ (IFN-γ), CCL3 and inducible nitric-oxide synthase (iNOS). Moreover, it is required for yielding the expression of chemoattractant chemokines (IL-8, macrophage inflammatory proteins 1 and 2, monocyte chemoattractant protein 1), matrix metalloproteinase 9 and several adhesion molecules (intercellular adhesion molecule 1, vascular cell adhesion molecule, P-selectin, E-selectin and mucosal addressin cell adhesion molecule 1). Accordingly, PARP enzymatic inhibition or PARP-1 gene knock causes the inhibition of cell migration to inflammatory sites [186]. Additionally, small molecule inhibitors acting on PARP reduced proinflammatory responses or enhanced anti-inflammatory functions of macrophages [187,188]. Nonetheless, it is involved in gene expression and the activation of neutrophils, macrophages, dendritic cells, microglia and other cell types [189,190,191]. It also influences the maturation and function of dendritic cells by regulating the production of IL-10 and IL-12 and the expression of costimulatory molecules CD86 and CD83 is fundamental for proper T cell activation and proliferation [192].

Accordingly, PARP1 seems to modulate T cells development and the differentiation of peripheral T cells into effector T cells such as T helper 1 (Th1), Th2, and Tregs [193,194,195]. As well, recent data showed a role in B cell development; mice with dual, but not individual, PARP-1 and PARP-2 deficiency exhibit a reduced number of B cells in the bone marrow, probably related to unrepaired DNA damage in proliferating B cells and not to an altered Ig V(D)J gene recombination [196].

Furthermore, PARP-1 can promote inflammation via PARylating of the high mobility group box 1 protein (HMGB1), that translocates to the cytosol and leak out of necrotic cells; once in the microenvironment, HMGB1 acts as a danger pro-inflammatory factor, reducing the clearance of apoptotic cells and fueling inflammation [197,198]. All these data provide evidences that PARP-1 and poly ADP-ribosylation can be important in regulating immunity and underscores the relevance of PARP-1 inhibitor development for immune disorders.

## 7. Concluding Remarks and Perspectives

Genome integrity and safety from potential invaders or endogenous threats are the main achievements of DNA repair systems and immune defences, respectively. However, these two pillars are clearly intimately interconnected, revealing a sustained, and sometimes, close coordination. The indefatigable work that has taken place over the years in both fields has unveiled important findings that enlighten common features and links between DDR and the immune system, and nevertheless, left many questions and unresolved riddles. Evidently, much work needs to be done to understand, at a deeper and detailed level, the overlapping pathways, and not least, to consider those discoveries from a further perspective than the cellular level, looking instead at the complexity of the organism as a whole. Indeed, the challenge remains as to what extent such mechanisms can be used to target cells with aberrant genomes during cancer, senescence or other pathologies. Currently, the -*omics* approach that is reasonably expanding in science makes us hope for an interdisciplinary and broader comprehension of the entire human organism in health and in disease, although, in some cases, the unavailability of the most proper animal model or the narrowness to explore the question in vitro or ex vivo is not sufficient to achieve unequivocal answers.

In the light of the above, a greater collaboration between oncologists and immunologists and between experimental and clinical research needs to be encouraged, in order to broaden knowledge and improve treatment strategies.

## Figures and Tables

**Figure 1 ijms-21-07504-f001:**
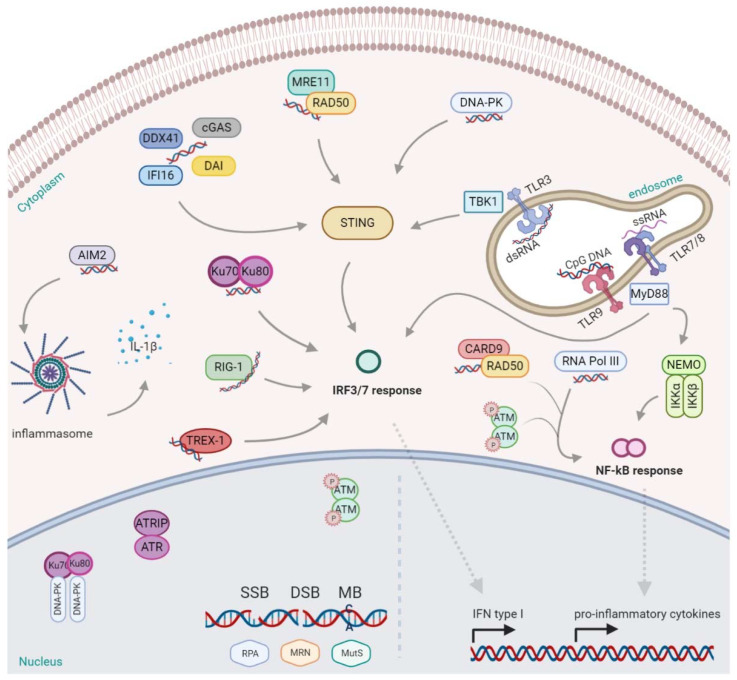
DNA sensing and activation of immune signaling.

**Table 1 ijms-21-07504-t001:** Representative causes of DNA damage and related DDR mechanisms.

Type of DNA Damage	Causes	DNA Repair Mechanisms	Mechanism Involved
Stalled replication forks, DSBs	Exposure to ionizing irradiation, UV, ROS or errors during DNA replication and replication-fork collapse	Homologous recombination (HR)	HR is largely restricted to S phase and G2 phase of the cell cycle, relies on the MRN complex, and repairs via double-strand break repair (DSBR) or synthesis-dependent strand annealing repair (SDSA). After incision, the 3′-end ssDNA coated with Replication Protein A (RPA) and Rad51 invades into a homologous DNA duplex. During DSBR, two Holliday junctions are formed, each between four strands of DNA that are then converted into recombination products. SDSA gives rise to non-crossover products. DNA polymerases fill in the gaps at the end of the invading DNA strand [30,31].
DSBs	Exposure to ionizing irrradiation, ultraviolet radiations (UV), ROS or errors during DNA replication and replication-fork collapse	Nonhomologous end joining (NHEJ)	NHEJ is initiated by the heterodimer of Ku70-Ku80 complex that recognizes and binds the broken DNA ends. The Ku70-Ku80 is an abundant nuclear complex and has high affinity for DNA ends that are either blunt or possess short ssDNA overhangs. To generate the two DNA blunt ends, this complex aligns the DNA ends, followed by (i) the activity of the DNA polymerases that fill in and (ii) the nucleases that trim off the DNA single-stranded overhangs. Then, the XRCC4/DNA ligase IV ligation complex is recruited to join the DNA ends together and promote end joining [30,32].
DSBs	UV, chemotherapy, ROS	Microhomology Mediated End Joining (MMEJ)	MMEJ includes three discrete steps, pre-annealing, annealing, and post-annealing of the microhomology (MH) flanking a DSB. PARP1 binds to DSB ends and facilitates the recruitment of resection factors [CtIP and Mre11 complex (Mre11/Rad50/Nbs1)] to expose MHs flanking DSBs. Those MHs that are placed far from the break usually require extensive resection by BLM/EXO1 to facilitate MMEJ. Annealing of MHs, which is inhibited by single strand binding RPA complex, induces the formation of non-homologous tails/flaps. These latter are then removed by XPF/ERCC1 nuclease before filling-in synthesis by Polθ and ligation by LigI/III [33].
Two nucleotide residues from opposite strands are covalently connected	Exogenous alkylating agents, cisplatin, mitomycin C or endogenous aldehydes, nitrous acid	Interstrand crosslink repair (ICL) or Fanconi Anemia (FA) repair complex	FA complementation group M protein detects DNA ICLs and induces the recruitment of the core FA complex at sites of damage. After the initial incision event, translesion DNA polymerases resume DNA replication in one strand and the resulting DNA DSB is processed by HR. In the G1 phase of the cell cycle, incision by ERCC1-XPF is followed by translesion DNA synthesis and the DNA ICL is looped out [34,35].
Nucleotide misincorporation	ROS and reactive nitrogen species (RNS) or endogenous problems during DNA replication leading to nucleotide misincorporation that creates base-base mismatches	Mismatch repair (MMR)	AG or TC mismatches are recognized by two heterodimers, MUTSa or MUTSb, that discriminate between the old and the newly synthesized strand, remove the mismatched nucleotide, and allow the replication machinery to use the original DNA template to restore the damaged DNA strand to its native form [36,37,38].
DSBs		Single Strand Annealing (SSA)	SSA involves annealing of homologous repeat sequences that flank a DSB, which causes a deletion rearrangement between the repeats. It is distinct from other HR pathways as it is independent from Rad51 recombinase and, instead, depends on Rad59 (which is indispensable to SSA when strand annealing is mediated by shorter (>30 bp) repeats). The successful annealing of repeat sequences forms unique recombination intermediates that contain one or two 3′ flaps; their cleavage is a key step in SSA as it produces DNA ends with the 3′ OH, suitable for repair synthesis by DNA polymerases. An endonuclease complex, XPF/ERCC1, catalyses the 3′ flap removal. Additionally, SSA requires proteins to stabilize the annealed intermediate and confer cleavage specificity [39].
Helix-distorting DNA lesions, base modifications, bulky adducts, intra-strand cross-links and thymidine dimers	UV, chemotherapy, ROS	Nucleotide excision repair (NER)	NER is divided into global genome NER (GG-NER) and TC-NER. In GG-NER, damage detection involves the XPC– RAD23B–Centrin2 complex. XPA, RPA, XPB, and XPD stabilize the damaged DNA and XPG and ERCC1-XPF structure-specific endonucleases cleave the 3′ and 5′ sides of the nucleotide fragment containing the damaged DNA. The single-strand gap is then filled by DNA polymerases and the nascent DNA fragment is sealed by DNA ligase III-XRCC1 and DNA ligase I [40]. Damage recognition in TC-NER involves the stalling of RNA polymerase II on the actively transcribed strand of a gene. RNA polymerase II is stabilized by the interaction with UVSSA, USP7 and CSB protein, which together recruit other factors like CSA. Once the remodeling of RNA polymerase II is completed, the TFIIH complex with XPA and RPA are recruited and that is where GG-NER and TC-NER converge [27,41].
Non-helix-distorting lesions. Base excision that leads to an AP site (apurinic/apyrimidinic site) when deoxyribose is cleaved from its nitrogenous base	Modification due to enzymatic activity, oxidation, deamination and alkylation; exposure to hydroxyl radicals that attack that weaken the glycosyl bond	Base excision repair (BER)	BER is a two-step process initiated by DNA glycosylases that detect and remove non-helix distorting DNA lesions through hydrolysis. The resulting abasic sites are cleaved by an apurinic/apyrimidinic endonuclease, exposing DNA SSBs that are repaired by either a short- or a long-patch repair mechanism depending on the number of replaced nucleotides. DNA ligase III and X-ray repair cross complementing protein 1 catalyses the nick-sealing step in short-patch BER, while DNA ligase I ligates the DNA SSB in long-patch BER. DNA polymerase b is typically involved during the DNA synthesis step [42].
Damaged base	Chemotherapeutic agents like dacarbazine and temozolomide	Direct repair (DR)	DR is the direct reversal of a damaged base to its native state without excision and de novo DNA synthesis. The DNA damage repaired in such way are of three types: photoreactivation by photolysases, O-methylation (in O^6^-Guanine, O^4^-Thymine and phosphates) by O^6^-methylguanine DNA methyltransferase (MGMT) and oxidative demethylation of N-methyl groups by AlkB family proteins. The self-methylated DNA methyltransferases are referred to as suicidal DNA repair proteins, as they are irreversibly inactivated during this stoichiometric repair reaction [28].

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
