# Peer review of "DNA Damage Response and Immune Defense"

_ijms, 2020, doi:10.3390/ijms21207504_

Round 1

Reviewer 1 Report

General comments:

The manuscript by Nastasi et al., provides a very interesting and detailed revision of the collected evidence on how an increasingly clear interplay between the DDR response and the immune system orchestrate the maintenance of genomic integrity. Moreover, they review existing evidence that the dysregulation of either system can upset this interplay and favor disease, namely autoimmune disease and cancer. Importantly, the authors also highlight how the growing understanding of this interplay is leading to the development of innovative approaches to improve treatment strategies. The manuscript is well structured and, for the most, well written. However, some issues require further attention.

Specific comments:

1-      Line 23- “common word”, perhaps replace by “common term”.

2-      Table 1:

  1. First row, 4th column – “After incision, the 3’-end ssDNA coated with Replication Protein A…”
  2. 6th row – SSA repairs DSBs, not SSBs as indicated in the 1st column.
  3. 7th row, 4th column – “…endonucleases cleave the 3’ and 5’ sides of the nucleotide fragment…”

3-      Line 129 – “When the innate…”;

4-      Line 186 – “In the absence of TLR9, other mechanisms…”

5-      Line 189 – “..mediated by the same or…”

6-      Line 204 – “…either viral in origin or generated by RNA polymerase III”

7-      Lines 215-216 – confusing sentence, please revise.

8-      Lines 220-223 – Two confusing sentences, please revise.

9-      Line 238 – “…thus, when tolerable DNA damage levels occur, the modulation or suppression of DDR signaling can mitigate…”

10-   Lines 242-265 – please arrange figure legend to match placing and font style.

11-   In contrast to the rest of the manuscript, topic 4 -“DNA damage inducers”, has several confusing and poorly constructed sentences. An extensive grammar revision is required for this topic.

12-   Line 498 – “… increasing the level of oxidative DNA damage and causing…”

13-   Line 540 – “ Despite the lack of solid evidence for efficacy of PARPi…”

14-   Line 561 – “First generation DNA-PKcs…”

15-   Lines 566-566 – “The repair machinery is able to fix DNA damages if cells have been prevented to go through their cycle progression. Thus, inhibition of cell cycle checkpoints like…”

16-   Line 567 – “toxicity as described for…”

17-   Line 578 – “… [43]; Nevertheless, cancers deficient in MMR frequently…”

18-   Line 568 – “… Germano et al. showed that it is possible to inactivate DNA…”

19-   Line 591 – “ Innovations in immunotherapies…”

20-   Lines593 -596 – “Active interventions improve the immune system response of the patient (for example, vaccination or adjuvant therapy) promoting antitumor effector mechanisms with cancer elimination and immune cell death (ICD) [12,13,167]. Passive interventions involve the administration of specific monoclonal antibodies (mAbs) against different immune checkpoints as well as adoptive transfer of genetically modified specific T cells called Chimeric Antigen Receptor T598 cell therapies (CAR-T).”

21-   Lines 607-608 – “PARPs have been studied as potential targets for drug development, with PARP1, and more recently PARP14, attracting most attention in the cancer field [173,174]”

22-   Line 620 – “By means of…”

23-   Line 621 – “…sensitive to other kinds of…”

24-   Line 638 – “…or enhanced anti-inflammatory…”

Author Response

Dear Reviewer 1,

we would like to thank you for the critics and comments that we received. Those ones guided us to improve the manuscript. As recommended, we edited all the mentioned misspellings and we revised and attempted to change the flow, not altering neither the structure nor the meaning, of paragraph 4.

Specific comments:

1-      Line 23- “common word”, perhaps replace by “common term”. DONE

2-      Table 1:

  1. First row, 4th column – “After incision, the 3’-end ssDNA coated with Replication Protein A…”
  2. 6th row – SSA repairs DSBs, not SSBs as indicated in the 1st column.
  3. 7th row, 4th column – “…endonucleases cleave the 3’ and 5’ sides of the nucleotide fragment…” ALL EDITED.

3-      Line 129 – “When the innate…”; NOW LINE 206, DONE.

4-      Line 186 – “In the absence of TLR9, other mechanisms…” NOW LINE 265, DONE.

5-      Line 189 – “..mediated by the same or…” NOW LINE 268, DONE

6-      Line 204 – “…either viral in origin or generated by RNA polymerase III”  NOW LINE 285, DONE.

7-      Lines 215-216 – confusing sentence, please revise. NOW LINES 295-300, we revised the sentences.

8-      Lines 220-223 – Two confusing sentences, please revise. NOW LINES 303-305, have been revised and edited.

9-      Line 238 – “…thus, when tolerable DNA damage levels occur, the modulation or suppression of DDR signaling can mitigate…” NOW LINE 321, DONE.

10-   Lines 242-265 – please arrange figure legend to match placing and font style. EDITED AND ARRANGED TO MATCHED STYLE.

11-   In contrast to the rest of the manuscript, topic 4 -“DNA damage inducers”, has several confusing and poorly constructed sentences. An extensive grammar revision is required for this topic.

WE ARE TRULY THANKFUL FOR THE SUGGESTION. WE READ THROUGH THE WHOLE PARAGRAPH AND WE REVISED THOSE UNCLEAR SENTENCES AND EDITED THE TEXT IN ORDER TO MAKE IT MORE READABLE, NOT ALTERING THE STRUCTURE NOR THE CONCEPTS. 

12-   Line 498 – “… increasing the level of oxidative DNA damage and causing…” NOW LINE 793. DONE.

13-   Line 540 – “ Despite the lack of solid evidence for efficacy of PARPi…” NOW LINE 835, DONE.

14-   Line 561 – “First generation DNA-PKcs…” NOW LINE 856, DONE.

15-   Lines 566-566 – “The repair machinery is able to fix DNA damages if cells have been prevented to go through their cycle progression. Thus, inhibition of cell cycle checkpoints like…” NOW LINE 861, DONE.

16-   Line 567 – “toxicity as described for…” NOW LINE 863, DONE.

17-   Line 578 – “… [43]; Nevertheless, cancers deficient in MMR frequently…” NOW LINE 877, DONE.

18-   Line 568 – “… Germano et al. showed that it is possible to inactivate DNA…” NOW LINE 885, DONE.

19-   Line 591 – “ Innovations in immunotherapies…” NOW LINE 890, DONE.

20-   Lines593 -596 – “Active interventions improve the immune system response of the patient (for example, vaccination or adjuvant therapy) promoting antitumor effector mechanisms with cancer elimination and immune cell death (ICD) [12,13,167]. Passive interventions involve theadministration of specific monoclonal antibodies (mAbs) against different immune checkpoints as well as adoptive transfer of genetically modified specific T cells called Chimeric Antigen Receptor T598 cell therapies (CAR-T).” NOW LINES 892-897, DONE.

21-   Lines 607-608 – “PARPs have been studied as potential targets for drug development,with PARP1, and more recently PARP14, attracting most attention in the cancer field[173,174]” NOW LINES 906-907, DONE

22-   Line 620 – “By means of…” NOW LINE 919, DONE.

23-   Line 621 – “…sensitive to other kinds of…” NOW LINE 920, DONE.

24-   Line 638 – “…or enhanced anti-inflammatory…” NOW LINE 943, DONE.

Reviewer 2 Report

In this manuscript, Nastasi et al reviewed the literature and presented the up-to-now knowledge on the link of DDR and immune responses. This association provides new perspectives for the investigation and management of immune-related diseases including cancer, and in this light the review contributes significantly to its field.

The manuscript is very well-written, easily readable with nice flow and organization. Data are well presented while nice table and picture are used.  The references reported are up-to-date..

I have no further comments/suggestions to add.

Author Response

Dear Reviewer 2,

We would like to thank you for the appreciation of our work. It is great to know it has been an easy read offering a new perspective for further investigations within the field. We tried to highlight the tight cross-talk between two different areas of research and we are glad to know it was appreciated.